# Automatic Segmentation of Ethnomusicological Field Recordings

**Matija Marolt \*** , **Ciril Bohak** , **Alenka Kavčič and Matevž Pesek**

Faculty of Computer and Information Science, University of Ljubljana, Večna pot 113, 1000 Ljubljana, Slovenia; ciril.bohak@fri.uni-lj.si (C.B.); alenka.kavcic@fri.uni-lj.si (A.K.); matevz.pesek@fri.uni-lj.si (M.P.)

\* Correspondence: matija.marolt@fri.uni-lj.si



**Featured Application: We describe a method for segmentation of ethnomusicological field recordings, which robustly segments field recordings into individual units labelled as speech, solo singing, choir singing, and instrumentals. We also present the SeFiRe segmentation tool that can be used to visualize and segment field recordings.**

**Abstract:** The article presents a method for segmentation of ethnomusicological field recordings. Field recordings are integral documents of folk music performances captured in the field, and typically contain performances, intertwined with interviews and commentaries. As these are live recordings, captured in non-ideal conditions, they usually contain significant background noise. We present a segmentation method that segments field recordings into individual units labelled as speech, solo singing, choir singing, and instrumentals. Classification is based on convolutional deep networks, and is augmented with a probabilistic approach for segmentation. We describe the dataset gathered for the task and the tools developed for gathering the reference annotations. We outline a deep network architecture based on residual modules for labelling short audio segments and compare it to the more standard feature based approaches, where an improvement in classification accuracy of over 10% was obtained. We also present the SeFiRe segmentation tool that incorporates the presented segmentation method.

**Keywords:** audio segmentation; field recordings; deep learning; music information retrieval

---

## 1. Introduction

Ethnomusicological field recordings are recordings gathered in the field, capturing folk music performances, usually intertwined with interviews with performers (informants) or commentaries by the folklorist. As their role is to document the legacy of folk musicians in their actual environment, the recordings are usually taken in non-ideal spaces (e.g., a performer's home) and may contain environmental noises (e.g., people entering and leaving the room, background talking) or interruptions. The practice of methodological gathering of field recordings started in the 1930s (by John and Alan Lomax), so ethnomusicological archives all over the world contain large recording archives spanning several decades. Older recordings were recorded with (by today's standard) poor recording equipment and may also suffer from signal degradations.

One of the first tasks that ethnomusicologists (or algorithms) face when studying a field recording, is its segmentation into smaller coherent units, such as units containing speech or individual folk song performances. Segmentation of audio recordings has a long research history starting with segmentation of speaking/non-speaking parts in speech recognition or segmentation of radio broadcasts into music/speech parts. Traditionally, segmentation approaches relied on hand-crafted features, tuned to discriminate between speech and music, which were the two categories most works aimed to

discriminate. Many approaches use standard timbral features, such as Mel Frequency Cepstral Coefficients [1] or chroma features [2], or develop new features, such as noise robust discriminators [3] or musically-informed Continuous Frequency Activation [4]. Based on the features, authors use different approaches to segment recordings, such as joint classification and segmentation by using a combination of standard hidden Markov models and multilayer perceptrons [5] or using a region growing algorithm to identify regions likely to contain speech or music followed by a maximum likelihood model that maximizes the probability of class labels given frame-level features and segment length limits [6].

Recently, as in other domains, deep neural networks are increasingly used for audio segmentation. In the Mirex 2015 music/speech classification and detection task [7], 11 authors submitted their classification algorithms, where three were based on deep neural networks, including the best system [8] (99.7% accuracy) that used one convolutional and one fully connected layer to classify log-Mel spectrograms as speech or music. Kruspe et al. [9] also explored deep neural networks for segmentation of broadcast signals, where they applied several fully connected layers over MFCC and chroma (CENS) features to discriminate between music/non-music, speech/non-speech, and noise/signal. Reported accuracies ranged from over 99% for silence/signal detection and 98% for speech/non-speech discrimination, down to 93% for music/non-music discrimination. In the Mirex 2018 music and speech detection task [10], all five authors used variants of different deep architectures for the segmenting TV broadcasts, where accuracies reached around 90% for speech and music detection. A more general audio classification approach was presented recently by authors from Google [11], who compared convolutional architectures for labelling audio from 70 M videos, where the best performing inception and residual networks achieved an AUC score of over 0.91 for classification of audio into 3000 classes. They also showed that features from the trained networks can be used to boost performance for other audio labelling tasks, such as predicting labels on the AudioSet dataset [12].

In this paper, we explore deep neural networks for labelling and segmenting ethnomusicological field recordings. Unlike broadcast recordings, field recordings are more challenging to label and segment due to their noisy nature. This was well demonstrated in the Mirex'15 music/speech segmentation task, which used a dataset that included several field recordings. The best detection system was based on hand-crafted features and a simple logistic regression classifier trained on field recordings [1]. It outperformed deep neural networks and achieved an average frame-based accuracy of 89%, 10% less than on the simpler clean speech/music discrimination task. We describe the dataset gathered for our experiments, which we make available to the community, the methods used for labelling and segmentation, and the results achieved, as well as the tools developed in the process.

## 2. Materials and Methods

The segmentation algorithm presented in this paper was designed to robustly label and segment ethnomusicological field recordings into consistent units, such as speech, sung, and instrumental parts. Resulting segmentations should be comparable to manual segmentations researchers make when studying recordings. Field recordings are documents of entire recording sessions and typically contain interviews with performers intertwined with actual performances. As these are live recordings of amateur folk musicians, they usually contain lots of "noise" and interruptions, such as silence when performers momentarily forget parts of songs, false starts and restarts, dancing noises, interruptions by other persons, or cars driving by. Performances may also change character; singing may become reciting, a second voice may join or drop out of a performance, etc.

The described nature of field recordings calls for a robust segmentation algorithm that would not over-segment a recording at each interruption—for example, we are not interested in each boundary separating speech and sung parts, as only some of them are actual segment boundaries. We would also like to distinguish between several different classes of segments and would like to take some prior knowledge of the classes into account. Last, we are not interested in millisecond-exact segment

boundaries or exact labeling of each small recording fragment; sometimes placing a boundary between two performances is a very soft decision and accuracy of a few seconds is good enough. Taking these points into account, we propose a three step approach to segmentation:

- First, a deep neural network is used to classify short audio segments into a set of predefined classes;
- then, a set of candidate segment boundaries is obtained by observing how the energy and class distribution change within the recording; and
- finally, the recording is segmented with a probabilistic model that maximizes the posterior probability of segments given the set of candidate segment boundaries with their probabilities and prior knowledge of lengths of segments belonging to different classes.

### 2.1. Deep Neural Networks for Labelling

Exploration of field recordings from a variety of ethnomusicological archives revealed four major classes of contents that appear in various cultures: Solo singing, choir (more than one singing voice) singing, instrumental performances, and speech. Our goal was, therefore, to classify field recordings into the four classes, and not to limit ourselves to just speech and music.

To train deep learning classifiers, large datasets are needed—the larger the better as recent experiences show. For speech/music segmentation, some datasets, such as the well-known GTZAN speech music collection [13], are available, but they mostly contain studio-grade samples labelled as speech or music, which is not very suitable for our goal. On the other hand, the recent Audio Set dataset [12] is an excellent large-scale audio classification dataset, however, its categories and contents are also not ideal for our purpose; for example, there is no solo singing category, examples labeled with singing are mostly accompanied by music, while the musical genres are mostly oriented towards popular music genres (pop, rock, etc.).

#### 2.1.1. Dataset

To train and evaluate deep learning models, we decided to gather and label a dataset that would contain short excerpts from a variety of ethnomusicological (and related) archives that put their collections online in recent years. The sources include: The British Library world & traditional music collection (https://sounds.bl.uk//World-and-traditional-music), Alan Lomax recordings (http://research.culturalequity.org/home-audio.jsp), sound archives of the CRNS (French National Centre for Scientific Research) (http://archives.crem-cnrs.fr/), and a number of recordings from the Slovenian sound archive Ethnomuse and the National Library of Norway, which are not available online, but were made available to us by ethnomusicologists with the respective institutions. These field recordings were augmented by the GTZAN music/speech collection, the Mirex 2015 music/speech detection public dataset, and the MUSAN corpus [14].

Five second long excerpts were extracted from recordings in these collections. To manually label them into the target classes, we enhanced the web-based audio annotator tool [15], so that it can be controlled exclusively by the keyboard. This makes labelling very fast when an excerpt contains just one class (e.g., speech). When an excerpt contains multiple classes, the user can still use the mouse to choose individual regions and label them accordingly. The enhancements enabled fast multi-user annotation of audio excerpts into the four main classes (speech, solo singing, choir singing, and instrumental), which we augmented with three additional classes. The "Not clear" class was introduced for sections containing too many short fragments of different types or for sections where the class is difficult to establish due to performance peculiarities (e.g., when it is difficult to discern between speech and singing). The "Noise" class was introduced for excerpts with no discernable contents, and, finally, the "Voice over instrumental" class to separately annotate this type of recording, which will be useful in our future work. The annotator's goal was to label each five second clip with the corresponding label(s), where the clips were randomly chosen from the dataset of unlabeled

clips for each participating annotator. The user interface of the annotation application, showing the spectrogram of an excerpt with the labelled class, was kept very similar to the original audio annotator and is shown in Figure 1.

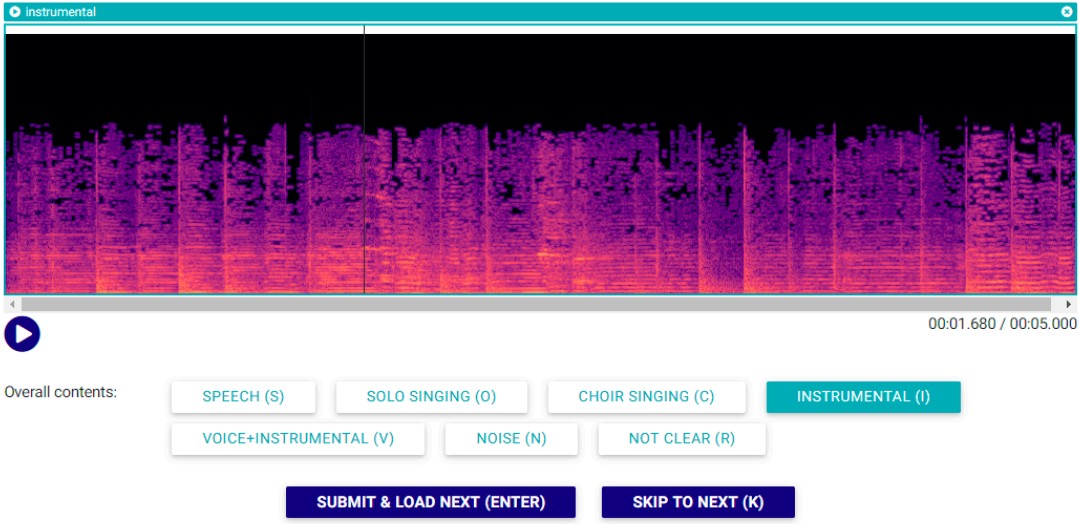

**Figure 1.** The annotation interface.

Using the interface, over 7000 excerpts were labelled. Approximately 80% of the excerpts were labelled into one of the four target classes, the distribution of class labels within the target classes was approximately equal (roughly 23% for speech, solo singing, and choir singing, and 33% for instrumentals). This was expected, as we chose the classes to be the ones most often represented in the field recordings. Ten percent of the other excerpts were labeled as vocals and instrumental, approximately 6% as not clear, and 4% as noise. The entire dataset with audio excerpts for internal sources and links to original materials is made publicly available with this paper (the URL is given at the end of the paper).

### 2.1.2. Network Architecture and Training

The basic architecture of the deep neural network for classification stems from our previous work [16], where we demonstrated their usability for the task. We chose convolutional deep networks as our main classification tool and focused specifically on residual networks [17], which demonstrated good performance for a variety of image, as well as audio-based tasks. The main feature of residual networks are their shortcut connections that implement identity mappings and enable convolutional blocks to learn residuals between the underlying mapping of features and the input. We augmented the first layer of the previously presented approach with music-specific feature filters and zero-mean convolutions, which together with data augmentation, improved the overall classification accuracy.

To provide network input, all the audio excerpts were first downsampled to 22,050 Hz, mixed to a single channel, and normalized. The audio was then split into 46 ms frames with a 14.3 ms step size, and 80 bin mel-scale spectra (30–8000 Hz) computed for all frames. We log-scaled the mel values, adding 1e-5 before applying the logarithm and used 2 s long feature blocks (80 $\times$ 140) as network inputs.

The network architecture is shown in Figure 2. The input is processed by two sets of music-specific feature filters: frequency filters and temporal filters. Frequency filters model changes in the frequency domain regardless of time—we used 10-by-1 filters, corresponding to the average frequency range of 700 cents. Temporal filters, on the other hand, model temporal dependencies independent of frequency—we used 1-by-10 filters, which thus correlate with approximately 143 ms (10 frames) of audio. This type of filter specialization has been shown to be effective for various audio processing

tasks, such as genre classification and emotion prediction [18,19]. Although each type of filters ignores one dimension (temporal or frequency), these dependencies are learned on higher network layers.

Additionally, to make the first layer filters more robust to varying recording conditions, we enforced the filters to have a zero mean, thus effectively learning to recognize the differences in the signal and ignoring constant offsets. Zero-mean convolutions were introduced to singing voice detection by Schlüter and Lehner [20] and were shown to be very robust to signal gain changes.

Output feature maps of the initial layer are subsampled by a 2 × 2 max pooling layer, and followed by four resnet v2 blocks [17], where the size of the feature maps is halved (in each dimension) and the number of filters doubles between three consecutive blocks. Exponential linear units [21] are used as activation functions of the resnet layers. The batch normalized output of resnet blocks is gathered by 1 × 1 convolutions into a 2D feature map. The map is processed by a small fully connected layer with four outputs, and the softmax activation function calculates final class probabilities.

The architecture and parameters of the higher layers are derived from our previous work [16], where we showed the efficiency of ELU and resnet layers for the task.

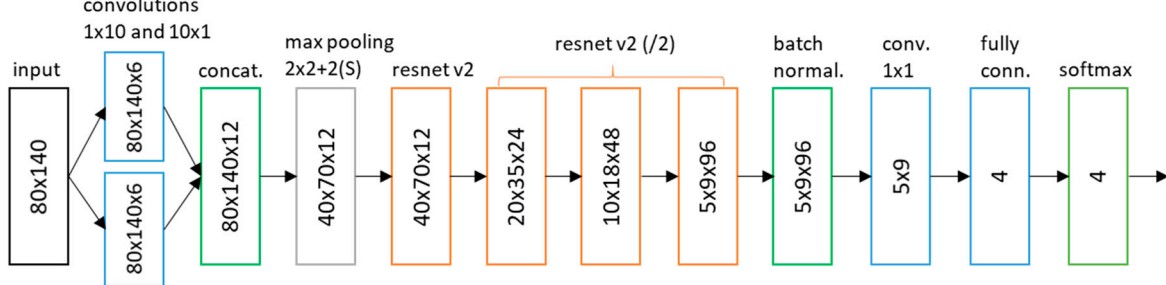

**Figure 2.** The network architecture.

Altogether the network contains 172,708 trainable parameters, which is relatively small in the deep learning world, where sizes of tens of millions of parameters are usual, we wanted to keep the network small due to the relatively small training set. Also, in our experiments, larger network sizes did not bring a significant increase in accuracy.

The network was trained on a dataset of 7136 audio excerpts with minibatches of 128 examples. For each audio example, the block of input features was drawn from a random location within the five second audio excerpt, so that for each epoch, the feature blocks used to train the network differed in their location within training files. Such time translation diversifies the limited training data available and improves performance, as was also demonstrated elsewhere [22]. In addition, time stretching and pitch transposition (scaling and shifting of the FFT spectrogram) of up to 30% was used to further augment the training data and reduce overfitting.

Stochastic gradient descent was used for training in 1000 epochs, and the learning rate was set to decay from 0.1 by a factor of 0.96 each 500 steps. The experiments were implemented in Tensorflow, and links to code are provided at the end of the paper. Training was done using the NVIDIA Quadro K5000 GPU and on average took 7 h for one model.

## 2.2. Segmentation

The goal of segmentation is to split the recording into a set of consistent units, such as speech or singing. Unlike broadcast segmentation, we are not interested in each small change of contents (e.g., just a few spoken sentences), but aim to segment the recording into larger units (entire songs or longer spoken parts). This is also where the presented approach differs the most when compared to other speech-music segmentation approaches (e.g., [5,6])—it is designed to segment audio into broader regions and to ignore small local changes.

The presented segmentation algorithm is a modification of our approach first presented in [1]. To segment a recording, we first find a set of candidate segment boundaries and calculate the likelihood

of splitting the recording at each boundary. We consider two criteria for boundary placement:
A criterion based on signal energy, such as when performances are separated by regions of silence
(or noise, since recordings may be quite noisy), and a criterion based on the change in signal content,
such as when speech is followed by singing. The way the two criteria are calculated is the main
distinction between the presented approach and [1].

To observe the presence of a signal (vs. silence or noise), we first calculate the RMS energy, $e$,
of the audio signal over 46 ms time frames. The energy is compared to two measures capturing the
global and local amplitude thresholds. First, the 1st percentile of the energy within 20 s time windows
is calculated and the global noise floor, $n_g$, is set to 5 dB over this value and lower bounded to $-60$ dB.
Such an adaptive estimate is needed, as field recordings are often noisy or recorded with varying
dynamics, so an absolute threshold would not be suitable as a global noise floor estimate. The local
noise floor, $n_l$, captures local dynamics and is set to 15 dB below the energy median filtered with a 6 s
time window. Based on both noise floors, the silence indicator function, $e_s(t)$, is set to a value of one,
when energy at time $t$ falls below any of the two noise floors, and zero elsewhere. The likelihood of
placing a boundary at time $t$, $p_s(t)$, is calculated by zero-phase filtering the indicator function with a
second order low pass filter with 0.1 Hz cutoff. An example is given in Figure 3.

Transitions between different kinds of signal content (e.g., speech to singing) are detected by
calculating the symmetric Kullback-Leibler (KL) divergence, $d(t)$, between probabilities of target classes
(as calculated by the deep classifier) within 10 s windows to the left and right of each time frame. KL
divergence will be large when the contents (class probabilities) on both sides will be different, and close
to zero, when they will be similar. The transition boundary placement likelihood, $p_t(t)$, is obtained
similarly to the silence likelihood by thresholding and low-pass filtering the divergence function, $d(t)$.
Both likelihood curves are depicted in Figure 3, which shows an excerpt of a field recording, where
content varies between speech and singing. The transition likelihood indicates segment boundaries due
to transitions between different content types well, while the silence likelihood indicates boundaries
between individual units separated by silence/noise, as well as short pauses, which often occur during
speech segments.

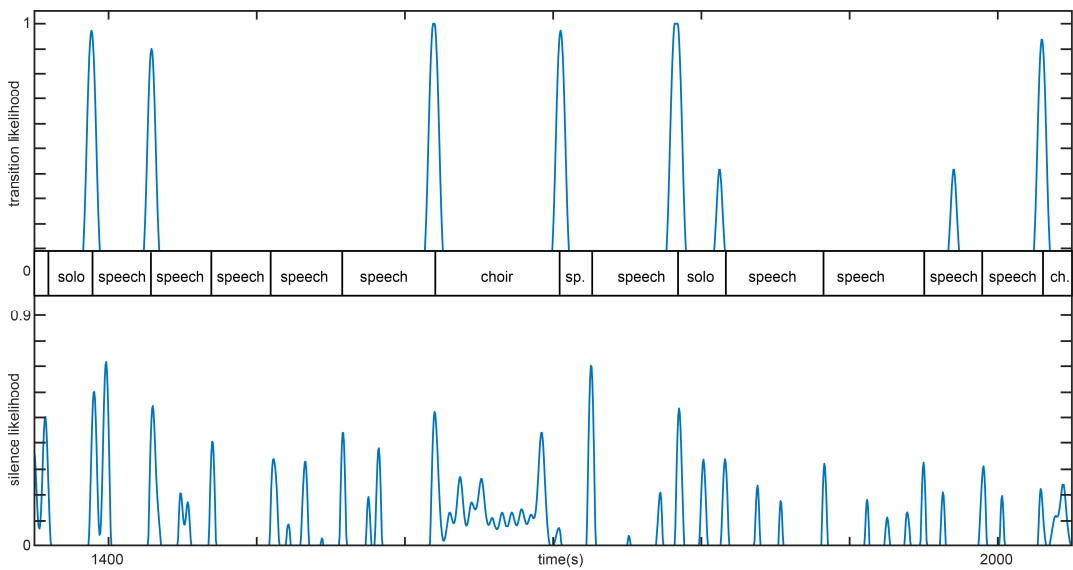

**Figure 3.** Segment boundaries and the two boundary likelihood curves: Transition and silence.

To segment a recording, we construct a probabilistic model, which integrates the data on segment
boundary likelihoods with prior knowledge of typical segment lengths. First, a set of boundary
candidates, $B_i$, is obtained by finding peaks of both curves. Based on the boundary candidates,
segmentation is defined as a sequence of segments, $S_{i1}, S_{i2}, \ldots, S_{iN}, 0 < i1 < i2 < \ldots < iN$, where

$S_{i1}$ starts at time 0 and ends at candidate boundary, $B_{i1}$, $S_{i2}$ starts at candidate boundary, $B_{i1}$, and ends at $B_{i2}$, $S_{i3}$ starts at $B_{i2}$ and ends at $B_{i3}$, and so on.

We treat each candidate boundary, $B_i$, as a discrete random variable with two outcomes: Either the candidate boundary represents an actual boundary and splits the recording into two segments, or not. The probability mass function for the variable is defined as the maximum of both likelihood curves:

$$P(B_i = true) = \max(p_s(t_i),\ p_t(t_i)). \tag{1}$$

In our model, the probability of each segment is only dependent on the location of the previous segment, so we can express the joint probability of all segments as:

$$P(S_{i1})P(S_{i2}|S_{i1})P(S_{i3}|S_{i2}) \ldots\ P(S_{iN}|S_{iN-1}). \tag{2}$$

To calculate the probability of placing a segment boundary at $S_i$ given $S_j$, we consider all candidate boundaries within the segment, as well as its duration. If the segment is to start at time $j$ and end at $i$, values of all candidate boundary variables within the segment must be *false*, while the value of the candidate boundary variable at time $i$ must be *true*. Segmentation can be further constrained by previous knowledge of the typical lengths of segments, $D_i$, given their class, leading to the following formulation:

$$P(S_i|S_j) = P(D_i|S_i,\ S_j)P(B_i = true) \prod_{j<k<i} P(B_j = false). \tag{3}$$

Probability, $P(D_i|S_i,\ S_j)$, of the segment duration given its boundaries is dependent on the class of the segment, as calculated by the deep classifier. Probability, $P(C_i = c|S_i,\ S_j)$, that the segment, $i$, belongs to class $c$ is calculated by averaging all class labels within the segment.

By analyzing durations of segments in our collection of field recordings, we estimated the means and standard deviations for all segment classes $(\mu_c,\ \sigma_c)$; for example, the duration of speech segments varies a lot and ranges from several seconds to over 10 min, while the average length of choir singing segments is around three minutes and their standard deviation below two minutes. By additionally enforcing minimal segment duration, $D_{min}$, we obtain the following expression for the probability of segment duration:

$$P(D_i|S_i,\ S_j) = \begin{cases} \sum_c P(C_i = c|S_i,\ S_j)G(i-j,\mu_c,\sigma_c) \\ 0, \qquad i-j < D_{min} \end{cases}, \tag{4}$$

where $G$ is the unscaled Gaussian function.

We find the sequence of segments that maximizes the joint probability as given in Equation (2) with dynamic programming, which leads to a simple and efficient solution. After segmentation is calculated, segments can be labeled by finding the class, $c$, that maximizes $P(C_i = c|S_i,\ S_j)$.

## 3. Results

In this section, we evaluate the presented labelling and segmentation algorithms on different datasets and compare their performance to other approaches.

### 3.1. Labelling

While there is a large number of published works dedicated to speech/music classification, they mostly deal with broadcast recordings, where their per-frame classification accuracies reach over 90% for classification into speech or music. Results may depend highly on the dataset, as recent Mirex music/speech detection results also show [7,10]. We are not aware of other works dealing specifically with field recordings, so to put our work into perspective, we compare it to three other approaches on the same dataset:

- A similar residual network architecture, but without music-specific filters in the input layer (12 3 × 3 filters were used), without zero-mean convolutions and data augmentation during training [16]. The network was trained on 64 channel mel spectrograms;
- a multilayer perceptron with one hidden layer of 16 neurons trained on the VGGish [11] features extracted from the data. VGGish are audio classification features extracted from a VGG-like deep model trained on a large YouTube dataset and made available by Google. Input to the MLP consisted of two consecutive 128-dimensional VGGish vectors, each summarizing 1 s of audio; and
- a simple logistic regression model trained on hand-crafted features, as described in [1].

Three-fold cross validation was used to assess the performance of each approach, where two thirds of the dataset was used for training, the remaining third for testing, and the procedure repeated three times. The train/test split was done on audio files, so the audio from the test set never appeared in the training data. Average classification accuracies and the number of parameters of each approach are listed in Table 1. The proposed model outperforms the others. It has approximately the same number of parameters as our previous approach; however, the music specific filters with zero-mean convolutions, combined with data augmentation, increased its accuracy. Both approaches use batch normalization and l2 regularization during training to avoid overfitting. In addition, a third of the dataset is used for testing at each run, so it is safe to assume that the performance is realistic for a wide variety of materials. The small neural network based on the VGGish features also performed well, and analysis showed that this approach makes a higher number of errors when discerning solo singing from speech, as well as choir singing from solo. These classes are likely underrepresented in the AudioSet dataset, so the features do not capture the distinctions well enough. Our previous approach based on a small set of carefully chosen features and a simple logistic regression classifier still achieves an almost 80% accuracy. As it is fast to calculate, it represents a viable option for classification in scenarios where speed is important and GPU not available (it is approximately seven times faster than the deep learning approach on a consumer grade GPU).

**Table 1.** Average classification accuracies for the compared approaches.

| Model | Number of Parameters | Accuracy |
|---|---|---|
| proposed approach | 172,708 | 0.92 |
| standard resnet | 172,936 | 0.89 |
| MLP on VGGish | 4180 | 0.8677 |
| logistic regression | 51 | 0.7958 |

An analysis of errors showed that many mistakes are logical and can be attributed to several factors. First, some of the recordings are very noisy and even a human listener can have some difficulty discerning the contents. Such recordings are often mistakenly classified as instrumentals, as the noise is considered part of the performance. The confusion matrix in Table 2 shows that many mistakes are made between neighboring classes: Solo singing is misclassified as choir singing or speech, choir mostly as solo, instrumentals as choir, or speech as solo. Some confusions may be due to the particularity of the contents, e.g., some short excerpts of dialectal speech may sound very much like singing, when old people sing, they may sound similar to speech. Chanting is also problematic, as it borders on speech and singing, while choir parts sung in unison, labelled as choir singing in our dataset, may sound very similar to solo singing and are often confused as such.

Some mistakes are not really mistakes—an excerpt may be correctly classified, and wrongly labelled. Namely, each audio clip in our dataset is only labelled with a single class, even though parts of it may contain another class. An example is a choir recording, where some parts are sung solo and then evolve into choirs. As the network only classifies short 2 s excerpts, it may correctly label the solo part as solo, however, the entire example is labelled as choir, so this is considered a misclassification.

If we consider the labeling of whole five second audio clips by averaging class probabilities within the clip and taking the maximum value as its class, the overall classification accuracy increases to 94%.

**Table 2.** The confusion matrix calculated from test results of all three cross validation folds.

|  |  | Predicted | | | |
|---|---|---|---|---|---|
|  |  | *solo* | *choir* | *instr.* | *speech* |
| **true** | *solo* | 0.84 | 0.07 | 0.02 | 0.07 |
|  | *choir* | 0.05 | 0.91 | 0.03 | 0.01 |
|  | *instr.* | 0.01 | 0.02 | 0.96 | 0.01 |
|  | *speech* | 0.05 | 0.01 | 0.01 | 0.93 |

*3.2. Segmentation*

We evaluated our segmentation algorithm on a set of 160 field recordings from the Ethnomuse archive, which were manually segmented by ethnomusicologists and contained 3703 segments. When evaluating the accuracy of our segmentation approach on this dataset, we needed to consider the goal of segmentation, which is not to separate every small excerpt of a particular type (e.g., short speech between units), but larger units, such as songs. This also means that boundaries in manual annotations are not placed very precisely, but often quite arbitrarily. For example, if two songs that follow each other are separated by an 8 s break or a short spoken sentence, the boundary annotation may be placed anywhere within the pause. To account for that and provide a realistic measure of the algorithm's performance, we needed to use wide evaluation windows (up to 8 s) within which the boundaries were counted as correctly placed—much larger as is customary, e.g., in Mirex evaluations, where 0.5 s and 1 s windows are used. Note, since the average segment length is over 120 s, such large windows sizes are still acceptable. In Table 3, we provide results of our approach with two differently sized windows: 3 s and 8 s. We also provide a comparison of the presented probabilistic approach to a baseline based on thresholding the silence and/or transition likelihood curves at the value of 0.5.

With larger window sizes, the algorithm can find a majority of boundaries (recall is at 76%). Precision is not so good, which means that the algorithm oversegments certain regions. This mostly occurs in long speech sections, which contain regions of silence that, for example, occur when people reflect on past events (consequently causing new boundaries), or in solo singing performances that are interleaved with reciting or spoken statements, such as "this is repeated three times and we start dancing in a circle so and so ..." (causing high KL divergence and new boundaries). False negatives occur when performances follow each other without significant changes, for example, several songs sung in a row almost without interruptions or when the boundary was placed because of the change of topic in speech parts, which the algorithm is not designed to detect. Segment start or end points may also be missed, because they interleave with speech, so that the boundary is placed either too soon or too late in a recording.

The accuracy of classification of correctly found segments into one of the four classes is similar to the overall classification accuracy of entire excerpts as presented previously—95%, errors are also similar to the ones described previously.

**Table 3.** Precision, recall, and F1 measure for segmentation with different tolerance windows.

| Algorithm | 3s Window | | | 8s Window | | |
|---|---|---|---|---|---|---|
|  | **P** | **R** | **F1** | **P** | **R** | **F1** |
| proposed approach | 0.47 | 0.62 | 0.52 | 0.56 | 0.76 | 0.63 |
| silence threshold | 0.51 | 0.38 | 0.40 | 0.58 | 0.44 | 0.46 |
| transition threshold | 0.45 | 0.42 | 0.40 | 0.56 | 0.54 | 0.51 |
| silence & transition thr. | 0.44 | 0.60 | 0.48 | 0.52 | 0.70 | 0.57 |

## 4. Discussion

Although it seems that with the advent of deep learning, which is already solving many difficult classification problems, music-speech classification and segmentation should be a relatively trivial task, recent work shows that we are still quite far from a perfect solution. Audio materials can be very diverse, so developing a robust approach that would process broadcast recordings on one side, and field recordings, on the other side of the quality spectrum, is difficult. The type of music the algorithms are trained on—e.g., commercial music vs. folk music, is also important, as is the goal of segmentation. Segmenting broadcast recordings, where the goal may be to detect all broadcast music, even short clips in the presence of speech (as in the recent Mirex evaluation [10]), is a different task to segmenting field recordings, where division into broader units is required, and short local changes need to be largely ignored. Also, as we show in our analysis of a wide spectrum of field recordings, these may require a finer level of classification than simply speech and music.

Therefore, we believe that research into specialized methods, such as ours, which is specifically targeted to segmentation of field recordings, is still needed. To disseminate our work, we make the dataset used for training our models, as well as the code for training and segmentation, available to the research community. To assist ethnomusicologists with their work, we also developed a segmentation tool, SeFiRe, which we make available to interested researchers. SeFiRe incorporates our classification and segmentation algorithms (as presented in this paper, as well as a music-speech segmenter [23]) to visualize and automatically segment field recordings according to their contents. The segmentation can also be manually corrected and exported.

SeFiRe visualizes field recordings as color-coded waveforms, where the color matches the contents. The four classes, speech, solo singing, choir singing, and instrumental performances, are mapped into: Dark blue, light blue, green, and red color. The display color of each short audio excerpt is interpolated between these basic colors according to the probability that the excerpt belongs to the respective class. Thus, where the classification algorithm is uncertain between two classes (e.g., solo vs. choir singing), the color reflects the uncertainty and is placed somewhere between the two.

The user interface of SeFiRe is presented in Figure 4. The user may load a new recording, play back and navigate the recording, and perform segmentation, whereby they can choose the algorithm as well as the sensitivity of segmentation, which influences the number of placed boundaries. The user may also manually correct, annotate, and store the resulting segmentation.

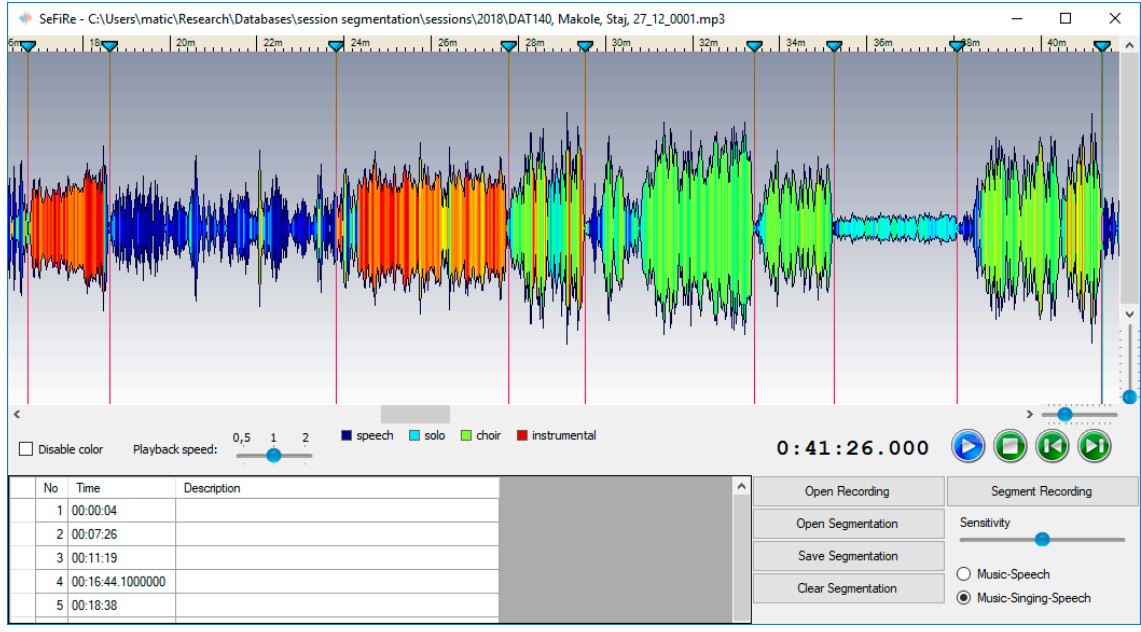

**Figure 4.** The SeFiRe (Supplementary Materials) user interface.

## 5. Conclusions

In the paper, we presented a method for segmentation of field recordings into individual units. Our method is composed of two main components: A deep neural network for classifying audio excerpts into four content types (speech, solo singing, choir singing, and instrumentals), and a probabilistic segmentation algorithm based on the detection of changes in signal energy and content. We presented an evaluation of our method, which showed that it performs favorably to the compared approaches, and described our tool for segmentation of field recordings, SeFiRe, which provides a user-friendly way for the interested public to use the presented algorithms for segmentation. The dataset, code for training the models, the trained model, and the SeFiRe tool are made publicly available with this paper.

Our future work will be directed primarily towards improving the classification. On the one hand, we wish to improve its accuracy by expanding the dataset and adding other external sources, such as the AudioSet, as well as using the weakly labelled paradigm for training. We also plan to expand on the number of classes, in the first place to include the voice+instrumental category, in order to be able to identify all parts of the field recordings that contain vocals. We will use a separate voice detector for the task. Furthermore, we plan to work on a hierarchical classification framework for the identification of instrument families and experiment with an integrated deep approach that would perform classification and segmentation jointly.

**Supplementary Materials:** The SeFiRe dataset is available online at: https://github.com/matijama/field-recording-db. The training code and the trained model are available online at: https://github.com/matijama/field-recording-segmentation. The SeFiRe segmentation tool is available online at http://lgm.fri.uni-lj.si/portfolio-view/sefire/.

**Author Contributions:** M.M. and M.P. conceived of and designed the experiments. C.K. performed the experiments and A.K. analysed the data. M.M., C.B., A.K. and M.P. wrote the paper.

**Funding:** This research was partly funded by the Slovenian Research Agency, within the project Thinking Folklore, grant number J7-9426.

**Conflicts of Interest:** The authors declare no conflict of interest.

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
