# Peer review of "Automatic Segmentation of Ethnomusicological Field Recordings"

_applsci, doi:10.3390/app9030439_

Reviewer 1 Report

This paper describe a method and software tool for the segmentation of ethnomusicological field recordings. The tool is intended to segments these recordings into sections of speech, solo singing, choir singing and instrumentals. The algorithm is based on convolutional deep networks, and is then uses a probabilistic technique for the segmentation. The dataset used comes from a number of sources. Its performance was compared with three other typical algorithms. It performed best in terms of accuracy with a figure of 92% but it was acknowledged that it is computationally intensive. The errors in the classification were analysed and the sources of some of the problems were identified. The segmentation algorithm did not work as well and again the errors were investigated. The software tool presented in the discussion section is named SeFiRe. It is flexible for visualizing and segmenting records. It is available on github along with the algorithms and test data.

Overall, this paper is very readable and well put together. It is clear what they are trying to achieve and how it fits in the field among the other references. This is actually well taken care of in the experimental design. The technical details are mostly described clearly. The results appear very reasonable. It is honest too to say that it is better but there is a computational burden. It is worthwhile that the sources of algorithmic errors are explained. The conclusions and future work are good. There are just a few points I think that would improve the paper:

On line 232 I think there should be italics where you have 0

It took a little while for me to understand Figure 2. I could link it up with the text with a little effort. Can it be improved with sacrificing its brevity? If not that is fine.

Coming into the results section it would benefit from a short paragraph at the beginning before 3.1 summarizing what is coming ahead. I was a bit confused when I read this section first.

line 280 please use the full version of approximately

lines 349 and 350 the sentence here needs to be looked over

Is there another paper on SeFiRe?

Can you say what the benefit of the Audio set is in the future work? 

Author Response

We thank the reviewer for the positive feedback. We implemented all desired corrections in the text, and slightly modified Figure 2, so we hope it's now more readable.

As far as the last two questions go: 

- SeFiRe was so far not presented in another paper. 

- The AudioSet will increase the variety of materials used for training the deep networks (although we use over 7000 excerpts, this is still a small dataset for deep learning), which hopefully will increase the accuracy further. We will however have to be careful to not skew the distribution of the classes too much (e.g. there is no solo singing in the AudioSet), so we plan to use the teacher-learner paradigm to bootstrap training.

Reviewer 2 Report

# Overall comments:

This is a well-written and enjoyable paper that makes a useful contribution in the application of deep learning and MIR to ethnomusicology.

The specific problems of audio classification and segmentation for field recordings are presented convincingly and justifies the development of new approaches.

The paper attacks two problems: classifying audio-chunks from long field recordings, and segmenting files based on these classifications. The classification problem has a sound ML approach using a convolutional neural network and evaluates well compared to previous approaches. This approach is documented well and should allow reproduction by other researchers.

The segmentation part appears to work sufficiently well for the application, but is not contextualised as well in terms of citing previous approaches or choice of benchmarks for the evaluation. Perhaps some more context for this approach could be provided.

The SeFiRe application is described in the discussion and provides good context for the application of this method. I'm not sure if this is intended to be a contribution of paper but it is useful to see that the proposed method fits into an existing application.

The paper seems to lack a conclusion with a summary of the main contributions and results. I think the authors should add this in order to help other researchers survey the field more efficiently.

I have included some specific comments and suggestions below.

## Specific Comments

S2, line 94: the three step approach could be turned into a list of steps in order to improve clarity.

S2.1.1 line 116: the full correct names of other institutions should be listed (e.g., CNRS (French National Centre for Scientific Research), and the National Library of Norway).

S2.1.2. This section is clear overall, but I was a bit confused about how chunks of audio were sent to the network. Is it correct that each example for the network is 2000ms long, divided in 140 46ms-long chunks with a step size of 14ms, each chunk yields 80 mel-scale spectra, and thus each example has size 80x140? I don't quite understand how the temporal filters end up processing 143ms of audio (line 166), perhaps some clarification could be provided here.

S2.1.2: training paragraph (lines 191-193): this paragraph does not say how long the network takes to train, how many examples were used in each epoch, and what computational resources were used. I think this is important information for reproducing this research. It is later stated that the network uses 172K parameters which could be stated here as well. Given the small size of the network (cf. 72M parameters for the VGGish net used in Section 3, it would be good to know how long this network takes to train, and why a bigger network is not necessary/desirable.

S2.2: Equation 3, line 246, D_i is not defined (but I guess it denotes duration).

S2.2: There's no context given for the segmentation method in the paper (e.g., citations to similar methods). Is this the same method (or a more developed version) as presented in [1]? It makes sense to me and appears to work, but I think it would be good to include information about how S2.2 fits into the other methods in the literature.

S3.1, line 277-278, how were the examples divided for cross validation? Given that each segment yielded multiple 5 second examples, were the sound files for testing kept separate from the training sound files? 

S3.1, line 280: suggest replacing "approx." with "approximate"

S3.1, line 285: suggest replacing with "VGGish also yielded good performance, but had problems.."

S3.1, line 290: first mention of "SeFiRe", but without context. I suggest just moving this sentence to the discussion where SeFiRe is more fully explained.

S3.1, line 290: The note about requiring a GPU does not make sense without information about the time and computational requirements for traing and inference, see previous comments.

S3.1 line 296, Table 2: what model/data is used to generate the confusion matrix?

S4, line 378, The AudioSet dataset is mentioned twice (also line 286), but not explained or cited in the paper.

S4, I feel that the discussion could have some comments about the size and training required for the classification network. This also relates to the 400x size difference between the proposed classifier (172K params) and VGGish (72M params). Are end users of SeFiRe expected to train new networks? If the network only need 172K parameters, it seems likely that they could, even without a GPU.

line 384: The paper is missing a conclusion. This may be a conscious decision by the authors, but it seems strange not to include a section summarising the main results and contributions of this work. My own impression was that paper ends somewhat abruptly without a conclusion section and that adding one would be beneficial. For this reason, I've marked "Conclusions are supported by results" as "must be improved".

line 425: reference [13] has no date.

Author Response

We thank the reviewer for the positive feedback and all the comments, which helped to improve the paper for the final submission. Below, we provide answers to the individual points raised.  

S2, line 94: the three step approach could be turned into a list of steps in order to improve clarity.

Implemented

S2.1.1 line 116: the full correct names of other institutions should be listed (e.g., CNRS (French National Centre for Scientific Research), and the National Library of Norway).

Implemented

S2.1.2. This section is clear overall, but I was a bit confused about how chunks of audio were sent to the network. Is it correct that each example for the network is 2000ms long, divided in 140 46ms-long chunks with a step size of 14ms, each chunk yields 80 mel-scale spectra, and thus each example has size 80x140? I don't quite understand how the temporal filters end up processing 143ms of audio (line 166), perhaps some clarification could be provided here.

We included a clarification and hope that this is now clearer. 1x10 filters process 10 consecutive frames, as these are 14.3 ms apart, this amounts to 143 ms if we disregard that the frame length is at 46 ms.

S2.1.2: training paragraph (lines 191-193): this paragraph does not say how long the network takes to train, how many examples were used in each epoch, and what computational resources were used. I think this is important information for reproducing this research. It is later stated that the network uses 172K parameters which could be stated here as well. Given the small size of the network (cf. 72M parameters for the VGGish net used in Section 3, it would be good to know how long this network takes to train, and why a bigger network is not necessary/desirable.

We included explanations on the time needed for training as well as why we chose not to scale the network.

S2.2: Equation 3, line 246, D_i is not defined (but I guess it denotes duration).

Implemented

S2.2: There's no context given for the segmentation method in the paper (e.g., citations to similar methods). Is this the same method (or a more developed version) as presented in [1]? It makes sense to me and appears to work, but I think it would be good to include information about how S2.2 fits into the other methods in the literature.

We described the distinction to [1] and included a statement to place the approach in the context of previous work.

S3.1, line 277-278, how were the examples divided for cross validation? Given that each segment yielded multiple 5 second examples, were the sound files for testing kept separate from the training sound files? 

They were divided based on files, so the test set was completely separate from the training. We included an explanation in the paper.

S3.1, line 280: suggest replacing "approx." with "approximate"

implemented

S3.1, line 285: suggest replacing with "VGGish also yielded good performance, but had problems.."

Implemented

S3.1, line 290: first mention of "SeFiRe", but without context. I suggest just moving this sentence to the discussion where SeFiRe is more fully explained.

Implemented

S3.1, line 290: The note about requiring a GPU does not make sense without information about the time and computational requirements for traing and inference, see previous comments.

We included a statement regarding the inference time.

S3.1 line 296, Table 2: what model/data is used to generate the confusion matrix?

We included an explanation in the table title

S4, line 378, The AudioSet dataset is mentioned twice (also line 286), but not explained or cited in the paper.

It was mentioned and referenced in the introduction, however the spelling was Audio Set, which we changed.

S4, I feel that the discussion could have some comments about the size and training required for the classification network. This also relates to the 400x size difference between the proposed classifier (172K params) and VGGish (72M params). Are end users of SeFiRe expected to train new networks? If the network only need 172K parameters, it seems likely that they could, even without a GPU.

In theory, anyone could train a network. The problem is of course in the data - we did our best to collect and label as many different sources as we could, but the procedure is time consuming. The more data, the more parameters a network may have. VGGish was trained on the huge AudioSet, so 72M parameters could be trained well. We have a much smaller dataset, so to avoid overfitting we need to be very careful with the network design. In our experience, less may be more, even  the 51 parameter logistic regression approach does not perform very poorly.

line 384: The paper is missing a conclusion. This may be a conscious decision by the authors, but it seems strange not to include a section summarising the main results and contributions of this work. My own impression was that paper ends somewhat abruptly without a conclusion section and that adding one would be beneficial. For this reason, I've marked "Conclusions are supported by results" as "must be improved".

We did not include the conclusion as the MDPI template and instructions do not really require it. However, we agree with the reviewer and included the section in the revised paper

line 425: reference [13] has no date.

This was actually a duplicate reference, which we removed.